# Spatial variation of the rain–snow temperature threshold across the Northern Hemisphere

Keith S. Jennings [1,2], Taylor S. Winchell[2,3], Ben Livneh[3,4] & Noah P. Molotch[1,2,5]

Despite the importance of precipitation phase to global hydroclimate simulations, many land surface models use spatially uniform air temperature thresholds to partition rain and snow. Here we show, through the analysis of a 29-year observational dataset ($n = 17.8$ million), that the air temperature at which rain and snow fall in equal frequency varies significantly across the Northern Hemisphere, averaging 1.0 °C and ranging from –0.4 to 2.4 °C for 95% of the stations. Continental climates generally exhibit the warmest rain–snow thresholds and maritime the coolest. Simulations show precipitation phase methods incorporating humidity perform better than air temperature-only methods, particularly at relative humidity values below saturation and air temperatures between 0.6 and 3.4 °C. We also present the first continuous Northern Hemisphere map of rain–snow thresholds, underlining the spatial variability of precipitation phase partitioning. These results suggest precipitation phase could be better predicted using humidity and air temperature in large-scale land surface model runs.

[1] Geography Department, University of Colorado Boulder, 260 UCB, Boulder, CO 80309, USA. [2] Institute of Arctic and Alpine Research, University of Colorado Boulder, 450 UCB, Boulder, CO 80309, USA. [3] Department of Civil, Environmental and Architectural Engineering, University of Colorado Boulder, 428 UCB, Boulder, CO 80309, USA. [4] Cooperative Institute for Research in Environmental Sciences, 216 UCB, Boulder, CO 80303, USA. [5] NASA Jet Propulsion Laboratory, 4800 Oak Grove Drive, Pasadena, CA 91109, USA. Correspondence and requests for materials should be addressed to K.S.J. (email: keith.jennings@colorado.edu)

Precipitation phase plays a critical role in the global hydrologic cycle and climate system, with snowfall and rainfall having divergent effects on land surface water and energy fluxes. Snow accumulation increases surface albedo, acting as a primary driver on the climate system[1], while winter snowpacks provide water storage for more than one billion people globally[2,3]. Climate warming has decreased the proportional amount of snowfall versus rainfall[4–6], reduced snow water equivalent (SWE) accumulation[5,7–10], shifted snowmelt earlier in spring[11–13], and diminished annual streamflow[14,15]. A greater proportion of future precipitation is predicted to fall as rain, further reducing snow accumulation in cold regions across the globe[16–21]. Climate warming is also predicted to increase the frequency and intensity of rain-on-snow events[22], which may significantly increase flood risks[23].

In this context, many land surface models (LSMs) estimate precipitation phase based on a simple, spatially uniform air temperature threshold and/or a range between two air temperatures in which a mix of rain and snow falls[24,25]. Incorrectly partitioning precipitation phase leads to significant biases in SWE, snow depth, and snow cover duration at both the point and basin scale[26–32]. These biases then propagate into errors in streamflow, land surface albedo, and surface–atmosphere energy exchange[26,29,30,33]. According to the Intergovernmental Panel on Climate Change, modeling the snow-albedo feedback—a function of snow cover extent and duration—represents a large source of uncertainty in LSM simulations of future hydroclimatic conditions[34]. There is therefore a need to critically analyze the way LSMs partition rain and snow.

Another method for predicting precipitation phase is through the application of atmospheric models with microphysics schemes that track a hydrometeor from its formation in the upper atmosphere to its deposition at the land surface[25]. Such an approach has been used to accurately simulate snowfall in several locations, including the Colorado Rocky Mountains[35] and the French Alps[26]. However, this manuscript focuses exclusively on methods that partition precipitation phase at the land surface due to the greater availability of surface forcing and validation data, the computational challenge of producing high-resolution, hemispherical-scale atmospheric model simulations, and the very wide use of LSMs using surface-based precipitation phase partitioning methods (greater than 2000 combined citations for the VIC and NOAH LSMs alone, according to the Web of Science). Furthermore, coarse-scale global circulation models (GCMs), such as those used in the Coupled Model Intercomparison Project[36], employ either surface or microphysics precipitation phase partitioning methods. Thus, a critical examination of rain–snow thresholds stands to benefit both the land surface and climate modeling communities.

Given the impact precipitation phase has on LSM output, it is essential that models accurately partition rain and snow. However, such a task is nontrivial, particularly at air temperatures near 0 °C[37]. Observational work indicates the temperature dependence of rain–snow partitioning follows a sigmoidal S-shaped curve with snowfall common above a surface air temperature ($T_s$) of 0 °C and increasingly less probable when approaching 4 °C[38–40]. Previous studies have developed rain–snow partitioning schemes based solely on $T_s$[41,42] or on $T_s$ plus near-surface humidity and/or air pressure[43–49]; yet, the broader applicability of these analyses is hindered by the limited spatial extent and range of conditions explored. In this regard, detailed analyses of phase partitioning—as well as its spatial variability and meteorological controls—over hemispherical scales have yet to be conducted.

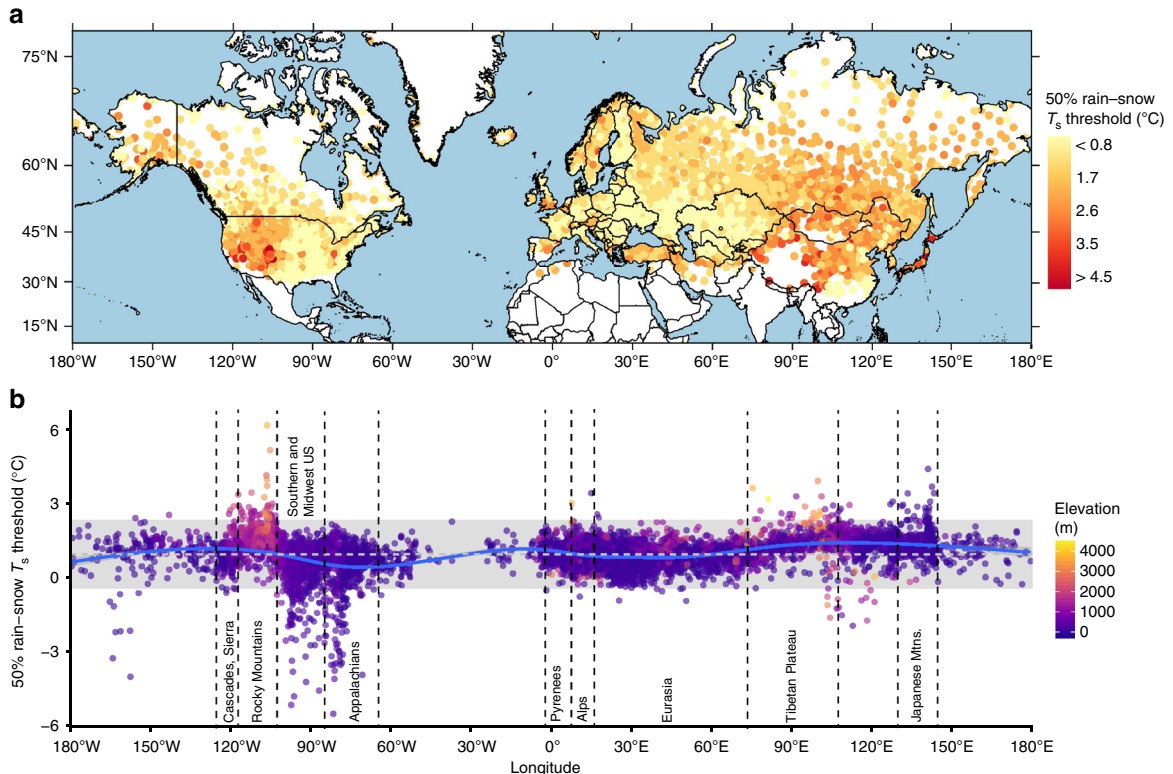

**Fig. 1** The observed 50% rain–snow $T_s$ threshold over the Northern Hemisphere for 6883 land stations from 1978 to 2007. Each point represents one station and only stations with a sufficient number of snowfall events were analyzed. **a** Thresholds mapped by station location. **b** Thresholds plotted by station longitude. The horizontal dashed line represents the Northern Hemisphere mean threshold (1.0 °C), the shaded gray box covers thresholds within ±2 standard deviations of the mean, and the blue line is a generalized additive model fit to the threshold data by longitude. Regions of interest are denoted by text within vertical dashed lines

A useful metric for defining the partitioning of precipitation phase at a given location is the 50% rain–snow $T_s$ threshold[42,48]. At this temperature precipitation occurs as rain and snow with equal frequency, while above the threshold precipitation is primarily rain and below primarily snow. The objectives of this study are to quantify the 50% rain–snow $T_s$ threshold over the Northern Hemisphere land surface, to assess how it varies with relative humidity (RH) and surface pressure ($P_s$), and to evaluate the impact of threshold selection on simulated snowfall frequency. We accomplish this through an analysis of a comprehensive 29-year (1978–2007) observational precipitation phase and meteorological dataset from 11,924 stations across the Northern Hemisphere ($n_{obs} = 17.8$ million), the application of a binary logistic regression phase prediction model using a spatially and temporally continuous reanalysis product in the Northern Hemisphere, and simulations of snowfall frequency using reanalysis data and a selection of precipitation phase methods. This study provides the most extensive empirical evaluation of precipitation phase over land, and the results have implications for predicting the response of rain–snow partitioning to climate change and discriminating between rain and snow in LSMs.

## Results

**Spatial variability of observed rain–snow thresholds**. Observed 50% rain–snow $T_s$ thresholds show marked spatial variation in the Northern Hemisphere (Fig. 1). We calculated an average threshold of 1.0 °C across 6883 stations (the remaining stations did not have enough data to fit the hyperbolic tangent as detailed in Methods), with 95% of observations falling in the range of −0.4 to 2.4 °C. Continental areas and mountain ranges generally exhibit the warmest thresholds, while maritime areas and lowlands exhibit the coolest thresholds. This is evident in the western United States, where values increase from approximately 0.6 to 1.5 °C near the Pacific Coast, Cascades, and Sierra Nevada to temperatures approaching 3.8 °C in the Intermountain West and Rocky Mountains. Thresholds east of the Rockies drop precipitously in areas influenced by the Gulf of Mexico, where rain is commonly observed below the freezing point.

In Europe, thresholds are generally near the Northern Hemisphere average of 1.0 °C, with higher values observed in the Pyrenees, Alps, and Scandinavian Mountains. Few observations in these longitudes are either extremely high or low ( ±2 standard deviations of the mean). Areas with weather patterns influenced by the Mediterranean, Black, and Caspian Seas exhibit some of the lowest thresholds in Eurasia. In Kazakhstan the threshold is typically less than 1.2 °C, except in upland areas near the Tien Shan mountains in the eastern portion of the country. Due to the low humidity of the region, central Asia—particularly the areas in and surrounding the Tibetan Plateau—consistently exhibits the highest observed thresholds, approaching 4.5 °C.

Generally, the highest 50% rain–snow $T_s$ thresholds are observed at upland elevations in continental regions, such as the Rocky Mountains and Tibetan Plateau. An exception is Japan, where thresholds are typically greater than the Northern Hemisphere average despite a maritime climate. This is largely attributable to synoptic-scale processes governing snowfall in the region, namely cold, continental air masses from Siberia acquiring heat and moisture from the Sea of Japan as they flow southeast. While in transit, the temperature lapse rate steepens, cloud top heights increase, and hydrometeors are formed in the masses' upper layers[50–52]. Surface observations, therefore, indicate markedly warmer conditions than where the hydrometeors formed aloft, while the high lapse rate reduces the probability that snow crystals will melt, giving Japan anomalously high 50% rain–snow $T_s$ thresholds. This unique synoptic setup

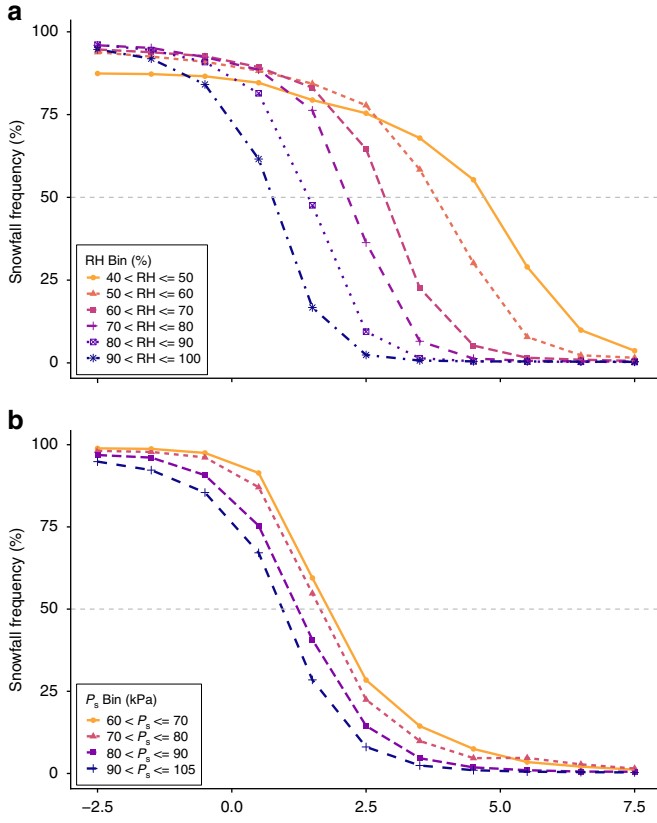

**Fig. 2** Snowfall frequency curves calculated using observations from 11,924 stations across the Northern Hemisphere (1978–2007). **a** Snowfall frequency curves plotted by RH bin. **b** Snowfall frequency curves plotted by $P_s$ bin. The 50% rain–snow $T_s$ threshold for each of the RH and $P_s$ bins are as follows: 4.5 °C (40–50%), 3.7 °C (50–60%), 2.8 °C (60–70%), 2.2 °C (70–80%), 1.4 °C (80–90%), 0.7 °C (90–100%), 1.9 °C (60–70 kPa); 1.7 °C (70–80 kPa), 1.3 °C (80–90 kPa), and 0.9 °C (90–105 kPa). Standard error bars are not plotted due to the large number of observations per RH, $P_s$, and $T_s$ bin (all standard errors are less than 0.9%). (The snowfall frequency curves for all Northern Hemisphere observations by $T_s$, $T_w$, and $T_d$ are available in Supplementary Fig. 1.)

underscores the inadequacy of partitioning precipitation phase with a uniform threshold in global applications.

**Meteorological controls on precipitation phase partitioning**. Based exclusively on the observational data, precipitation events that occur at low RH are more likely to fall as snow at higher $T_s$ than events coinciding with high RH (Fig. 2a). For example, the probability of precipitation falling as snow at 2.5 °C is over 30-times greater in the lowest RH bin compared to the highest. At 0.0 °C all snowfall frequency values are greater than 70% across the RH bins; however, as $T_s$ increases the curves exhibit a stark separation with the snowfall frequency of the higher RH curves dropping quickly toward zero, whereas the lower RH curves maintain greater snowfall frequency values at higher $T_s$. RH also exerts a strong control on the 50% rain–snow $T_s$ threshold, which ranges from 0.7 °C in the 90–100% RH bin to 4.5 °C in the 40–50% RH bin. Each 10% increase in RH is associated with a 0.8 °C decrease in the 50% rain–snow $T_s$ threshold. These findings are consistent with hydrometeor energy balance theory, in that a low ambient RH facilitates evaporative cooling through latent heat exchange, thus enabling a snow crystal to maintain its frozen state in an above-freezing atmosphere. Additionally, 50%

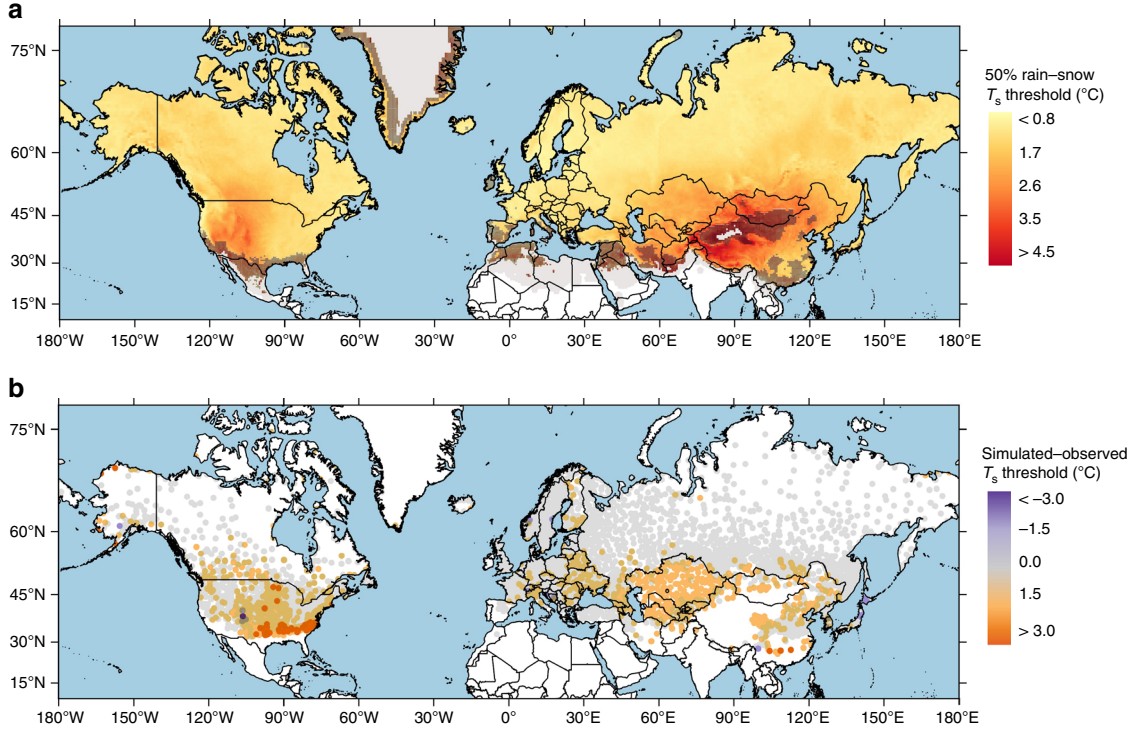

**Fig. 3** The simulated 50% rain–snow $T_s$ threshold across the Northern Hemisphere and the difference between simulated and observed thresholds. **a** The simulated 50% rain–snow $T_s$ threshold is computed using a hyperbolic tangent fit to simulations of precipitation phase using a binary logistic regression model applied to gridded MERRA-2 reanalysis data (see Methods). Hatching indicates there were not enough data to compute the threshold using a hyperbolic tangent and the resultant threshold was calculated using a linear regression based on snow probability between 0.5 and 6.5 °C. Areas shaded in gray had some modeled snowfall, but there were not enough total events and/or snowfall days per temperature bin in order to identify a 50% rain–snow $T_s$ threshold using either method. Note that this may occur for regions that are typically cold (e.g., central Greenland) or warm (e.g., northern Africa). Areas with no shading did not have any modeled snowfall in the −8 to 8 °C temperature range. **b** Differences between the simulated and observed 50% rain–snow $T_s$ thresholds. Gray shading indicates differences were within ±1 °C (81.0% of stations)

rain–snow wet bulb temperature ($T_w$) and dew point temperature ($T_d$) thresholds are colder than $T_s$ thresholds given that the former are lower relative to $T_s$ when conditions are unsaturated (Supplementary Fig. 1).

Observations indicate $P_s$ also influences the 50% rain–snow $T_s$ threshold, but to a lesser extent than RH (Fig. 2b). The threshold ranges from 0.9 °C in the 90–105 kPa bin to 1.9 °C in the 60–70 kPa bin, and each 10 kPa increase in $P_s$ is associated with a 0.3 °C decrease in the 50% rain–snow $T_s$ threshold. Thus, lower $P_s$ is associated with an increased probability of snowfall at a given $T_s$, indicating higher elevation sites are likely to see snowfall at a warmer $T_s$ than lower sites. The 1.0 °C spread in the 50% rain–snow $T_s$ thresholds between the highest and lowest $P_s$ classes is significantly lower than the 3.8 °C spread in the $T_s$ thresholds between the highest and lowest RH classes. These results indicate that RH, as opposed to $P_s$, is a greater determinant of the 50% rain–snow $T_s$ threshold and the probability of snowfall at a given temperature.

As noted in the introduction, some LSMs employ a temperature range in which rain and snow are proportionally allocated in order to represent mixed-phase precipitation events. The difference between the minimum and maximum $T_s$ values for these ranges is generally between 1.0 and 3.0 °C[25,42]. Although values of rain–snow proportions are not included in the observational dataset, we used the precipitation phase data to evaluate the temperature ranges in which rain and snow were probable for the different RH and $P_s$ bins. In this case, we considered the 90 and 10% rain–snow $T_s$ thresholds to define the minimum and maximum $T_s$ values for mixed-phase events, respectively. For this part of the analysis, the 90% threshold

represents the $T_s$ value at which 90% of observed precipitation falls as snow and 10% as rain, and vice versa for the 10% threshold. Similar to the 50% rain–snow $T_s$ thresholds, the 90% and 10% thresholds are warmer for lower RH and $P_s$ bins, meaning drier and higher sites consistently experience more snowfall at higher $T_s$ (Supplementary Table 1). In addition to the thresholds occurring at higher $T_s$, the ranges are also wider for the lower RH bins, indicating rain and snow are probable over a larger $T_s$ range for storm events with low RH. Overall, the computed $T_s$ range between the 90 and 10% thresholds has a minimum width of 2.6 °C and a maximum of 4.6 °C, indicating that LSMs may benefit from using wider $T_s$ range parameters when prescribing rain–snow proportions.

Furthermore, our finding of snowfall occurring at higher $T_s$ under dry ambient conditions, as presented above, stands in contrast to Dai[40], who found snowfall to be more likely over the ocean than over land at a given $T_s$ despite the higher humidity of marine environments. Dai[40] posited that this phenomenon was likely a function of the increased temperature lapse rate above the ocean marine layer, which leads to lower freezing levels and reduces the time a hydrometeor spends falling through the warm lower troposphere. The discrepancy may also arise from the two different analysis methods. In this regard, Dai[40] aggregated all land-based observations into one group, whereas our method bins the land stations into humidity and pressure classes and we did not quantify spatial variation in the 50% rain–snow $T_s$ threshold over the ocean. Additionally, Dai[40] suggested no pressure-phase relationship above 75 kPa, whereas our analysis showed clear divergence in the snowfall frequency curves at the examined pressure bins.

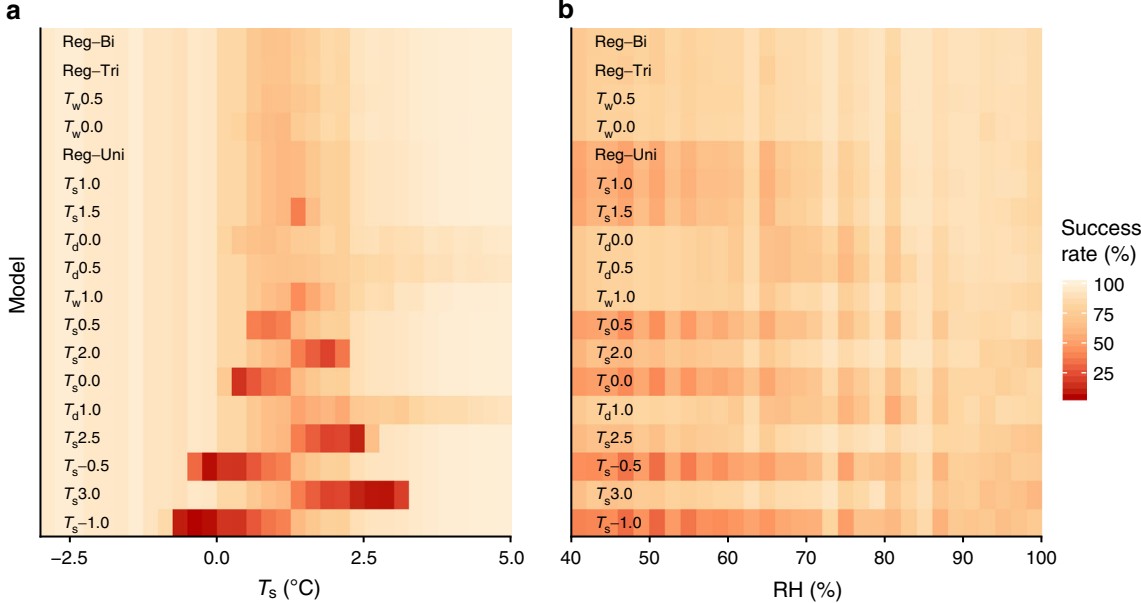

**Fig. 4** Success rates for the different precipitation phase methods ordered by average success rate with the best performing methods at the top. The success rate value is equal to the percentage of precipitation phase observations correctly simulated by the precipitation phase method. Low success rate values (dark red) correspond to a high frequency of misidentified precipitation phase observations. **a** Success rate plotted by $T_s$. **b** Success rate plotted by RH. Details on the different methods and average success rates are available in Supplementary Table 3

We also note the majority of observations were recorded in the higher RH (i.e., 90–100%) and $P_s$ (i.e., 90–105 kPa) classes. For example, records in the 90–100% RH bin outnumber those in the lowest three RH bins by more than an order of magnitude (12.7–1). For $P_s$, the distribution is similar in that observations in the 90–105 kPa bin outnumber those in the three other bins by more than 8.6–1. This sampling bias results from a combination of the increased probability of precipitation at higher RH and the greater representation of lower elevations in the observations (i.e., higher $P_s$). The latter introduces uncertainty into the results as the stations were not strategically located to cover the full range of hydrometeorological conditions in an unbiased manner. This uncertainty is a candidate for further research, particularly in mountainous regions where seasonally snow covered areas—vital to water resources—are located at high elevations with low $P_s$.

**Simulations of rain–snow partitioning.** Figure 3a presents a spatially continuous product simulating the 50% rain–snow $T_s$ threshold over land in the Northern Hemisphere as produced by a logistic regression model run on 27 years of MERRA-2 reanalysis data (see Methods). Here station precipitation phase observations were used to optimize a bivariate model with the predictor variables $T_s$ and RH. As with the station observations, the simulated 50% rain–snow $T_s$ threshold displays marked spatial variability with the highest values in the Rocky Mountains of North America and the Tibetan Plateau of central Asia. The lowest thresholds are generally simulated in areas with maritime climates such as the Pacific Northwest of the United States and northern Europe. Figure 3b displays the difference between the simulated threshold at each grid cell and the corresponding observed station threshold, where available. The simulated threshold is generally within ±1 °C of the observation and the mean bias is 0.5 °C across the Northern Hemisphere. Notwithstanding, prominent over-estimates of the $T_s$ threshold occurred in the southeastern United States and central Eurasia (Fig. 3b, red dots), while under-estimates were less common (Fig. 3b, purple dots). In addition to the low bias, model standard deviation (0.45

°C) is similar to the observations (0.68 °C) but variability is lower overall.

In order to produce the map presented above, we considered three binary logistic regression model versions: univariate (precipitation phase predicted by $T_s$); bivariate ($T_s$ and RH); and trivariate ($T_s$, RH, and $P_s$) (for optimized model coefficients, see Supplementary Table 2); in addition to a suite of 50% rain–snow $T_s$, $T_w$, and $T_d$ thresholds from the literature (Supplementary Table 3). Figure 4 displays the success rate of each method in predicting the precipitation phase of the validation data at a given $T_s$ (a) and RH (b). Notably, the top two methods are the optimized bivariate and trivariate regressions and the best four methods all incorporate humidity, either through RH or $T_w$. Conversely, the worst four methods rely on $T_s$ alone. Based on this evaluation, cool (−1.0° to 0.0 °C) and warm (2.0–3.0 °C) 50% rain–snow $T_s$ thresholds that under- or overpredict snowfall are clearly inappropriate for LSM runs over large spatial extents.

Each of the top 10 methods correctly predicts precipitation phase > 84.0% of the time across the examined $T_s$ and RH values. However, this includes periods when the $T_s$ would clearly indicate rain or snow. All methods, except for the worst eight thresholds, show a loss of skill between 0.6 and 3.8 °C, illustrating the difficulty in phase partitioning at $T_s$ near freezing. Within this range, the dip in performance for the best $T_s$ and RH method is less than that for the best $T_s$-only method. Specifically, the bivariate ($T_s$ and RH) model reaches a minimum success rate of 68.7%, which is a 13.1% improvement on the 60.7% success rate minimum for the univariate ($T_s$ only) model. The difference is even larger when examining the performance by RH, where the bivariate minimum success rate is 35.3% higher than that of the univariate model. Furthermore, the methods incorporating humidity provide consistent performance across the RH range, while the $T_s$ models exhibit downgraded performance at lower RH values. In this case, the $T_w$ thresholds have the lowest standard deviation in success rate (5.0%), while $T_s$ thresholds have the highest (15.0%). Therefore, including humidity—whether through RH or $T_w$—provides a marked improvement

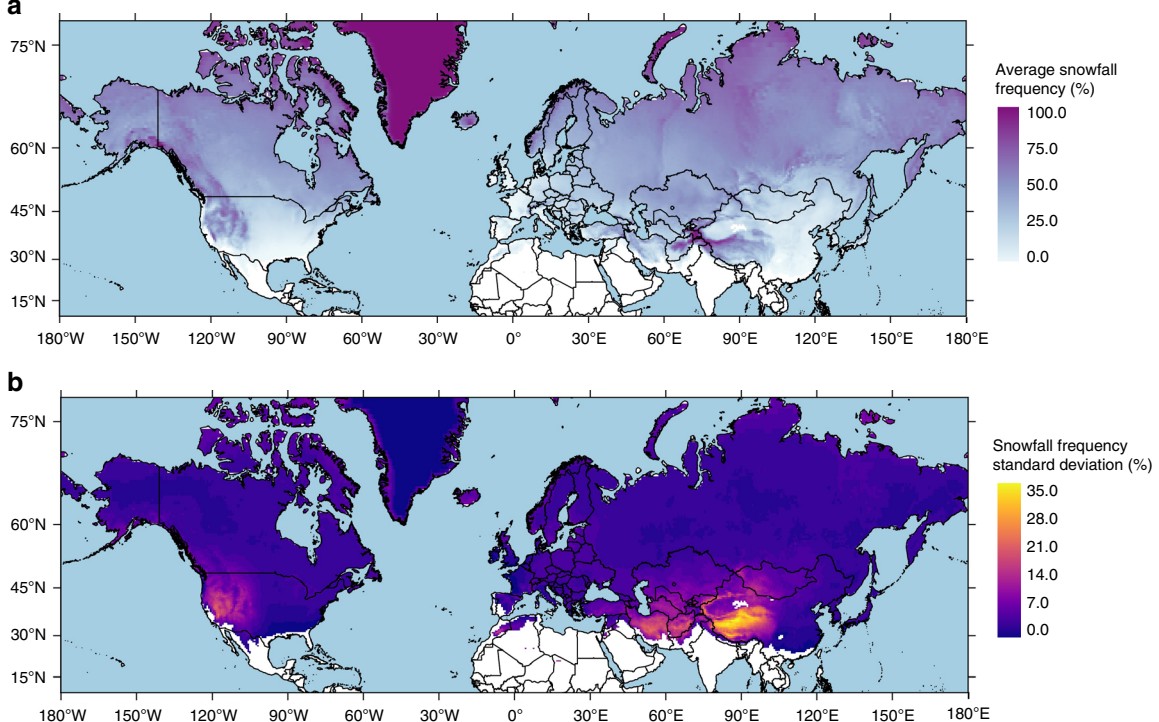

**Fig. 5** Simulated average snowfall frequency and its standard deviation across the Northern Hemisphere land surface as computed using 18 different precipitation phase methods applied to MERRA-2 reanalysis data. **a** Average snowfall frequency for the 18 methods. **b** Snowfall frequency standard deviation for the 18 methods. Only grid cells with an average of at least one snowfall event per year were included in the above maps. Land areas in white represent both hot (e.g., India and Southeast Asia) and arid regions (e.g., the Taklamakan Desert in northwest China and the Saharan Desert) where snowfall is rare

in skill when precipitation phase prediction is at its most uncertain.

**Snowfall frequency sensitivity to phase partitioning method.** In order to evaluate the impact of misdiagnosing precipitation phase on snowfall simulations, we computed the average and standard deviation of snowfall frequency across the Northern Hemisphere using 18 different precipitation phase methods (Supplementary Table 3) applied to 27 years MERRA-2 reanalysis data. Predictably, average simulated snowfall frequency increases with latitude and elevation, and is lowest in the tropics, hot deserts, and maritime regions (Fig. 5a). The standard deviation of snowfall frequency, an expression of uncertainty in this analysis, is generally < 10%, notwithstanding semiarid regions where it exceeds 30% in some cases (Fig. 5b). Precipitation phase partitioning is most sensitive to method choice in lower-humidity areas such as the Intermountain West of North America and the Tibetan Plateau of Asia, both of which rely heavily on snowpack for regional water resources[2,3,53] (standard deviation of snowfall frequencies are provided for major river basins in Supplementary Table 4). Harpold et al.[54] similarly noted that snowfall frequency was sensitive to the choice of a $T_s$-only versus a $T_s$ and RH precipitation phase method in arid and semiarid areas of the western United States. High-elevation, semiarid areas also show the greatest difference in snowfall frequency when comparing the trivariate ($T_s$, RH, $P_s$) to the bivariate ($T_s$, RH) phase regression model (Supplementary Fig. 2).

The 18 precipitation phase methods show a large spread in average Northern Hemisphere snowfall frequency (Fig. 6a) with $T_w$ and $T_d$ temperature methods producing larger snowfall frequencies relative to the $T_s$ methods. Figure 6b shows how the uncertainty produced by the different phase methods scales with

the annual simulated snowfall frequency. The lowest standard deviations are observed at snowfall frequencies near 0% and 100%, indicating the selection of the precipitation phase method produces little variability in locations that are currently either rain dominated or snow dominated. Most standard deviations are <10% (Fig. 6c) with the greatest values observed near an average snowfall frequency of 50% and an average $T_s$ of 0.0 °C (Fig. 6b), indicating temperate areas with a rain–snow mix are also sensitive to the phase method selection.

Climate warming is expected to reduce snowfall frequencies over many regions vital to water resources and the global climate system[2,54–58] with areas of average $T_s$ near 0 °C considered most at-risk to warming[59]. Our research suggests that modeling studies may be misrepresenting the amount of simulated future snowfall, primarily as a result of the application of spatially uniform $T_s$-based precipitation phase methods. For example, the variable infiltration capacity (VIC) macroscale model has been used for myriad studies in snow-dominated regions worldwide[3,7,55]. VIC, like many LSMs, employs a rain–snow $T_s$ range centered around a default 50% rain–snow $T_s$ threshold—in this case 0.0 °C—i.e., 1.0 °C cooler than the globally observed $T_s$ threshold, meaning the model will underpredict snowfall over large areas. Storck and Lettenmaier[60] found calibrating the VIC $T_s$ threshold improved wintertime SWE predictions while degrading spring SWE estimates, likely an effect of the inconsistent performance of $T_s$-only methods. Exacerbating the issue is that continued climate warming will cause cold and temperate regions to see a shift to $T_s$ near freezing[34], which will make predicting rain and snow in these areas more uncertain, particularly if inaccurate phase prediction methods are used.

In that regard, recent review papers have called for improvements in the way precipitation phase is represented within

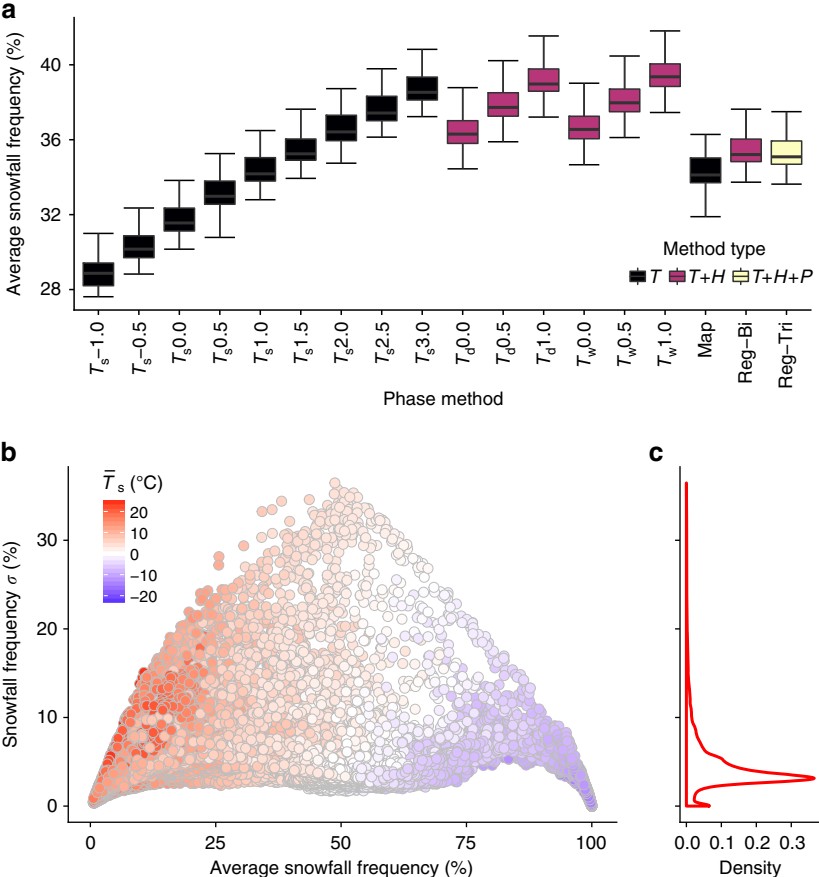

**Fig. 6** Average Northern Hemisphere snowfall frequency and standard deviation. **a** Average Northern Hemisphere snowfall frequency as computed by the 18 different methods applied to 27 years of MERRA-2 reanalysis data. Shading represents the data included in each method ($T$—air temperature; $H$—humidity via RH, $T_w$, or $T_d$; and $P$—pressure). **b** The standard deviation of snowfall frequency plotted against average snowfall frequency for each MERRA-2 grid cell for the 18 precipitation phase methods. Shading represents the average $T_s$ for that grid cell during precipitation events. **c** Density plot showing the distribution of snowfall frequency standard deviations

models across multiple Earth sciences disciplines[24,25]. Our work has shown, through intensive analysis of a hemispherical-scale precipitation phase dataset and a suite of snowfall frequency simulations, that the marked spatial variability in 50% rain–snow $T_s$ thresholds is primarily a function of RH and that methods incorporating RH are more effective at predicting precipitation phase. Thus, modelers should employ a precipitation phase method that represents physical processes and spatial variability, particularly when there are no observations of precipitation phase on which to calibrate a threshold parameter. Ultimately, these findings have broad implications for historical and future simulations of the hydrologic cycle, and for estimating the impacts of climate warming on snow accumulation, land surface albedo, streamflow, soil moisture, and land–atmosphere energy exchange.

## Methods

**Observational Data**. In this study, we analyzed observations from the National Center for Environmental Prediction (NCEP) Automated Data Processing (ADP) Operational Global Surface Observations dataset (DS464.0), hosted by the National Center for Atmospheric Research (http://rda.ucar.edu/datasets/ds464.0/). This dataset includes 6- and 3-hourly synoptic weather reports with measurements of $T_s$, dew point temperature ($T_d$), and $P_s$ (collected at ~1.5–2.0 m above ground), as well as visual observer reports of precipitation phase from meteorological stations across the globe for the period 1978-01-01 through 2007-02-25. While records were available from both land and ocean stations, we used data exclusively from land-based stations in this study. Stations in regions where precipitation falls exclusively as rain (e.g., the tropics) were not included in the analysis and we focused solely on the Northern Hemisphere due to its greater land surface area, larger seasonal snow

cover extent, and increased number of surface observations relative to the Southern Hemisphere.

We classified precipitation reports as either rain or snow using the World Meteorological Organization precipitation phase categories described in detail in Dai[40,61]. Precipitation amounts were not included in the dataset and we removed sleet as well as potential mixed-phase observations from the analysis because the relative proportions of solid and liquid precipitation during such events were not reported (i.e., it was impossible to quantify the amount of precipitation falling as snow versus rain). The classification of precipitation events was then used to quantify the rain–snow frequency per 1 °C $T_s$ bin from –8 to 8 °C at each station. In other words, if there were 100 total precipitation observations from 1 to 2 °C, 75 of which were snow, the snowfall frequency in that bin would be 75.0%. We then calculated the 50% rain–snow $T_s$ threshold for each station using the approach of Dai[40], where a sigmoidal curve is fit to observations of snowfall frequency per 1 °C $T_s$ bin from –8 to 8 °C using a hyperbolic tangent function:

$$T_{50} = \frac{\tanh^{-1}\left(\frac{F}{a} + d\right)}{b} + c \qquad (1)$$

where $T_{50}$ equals the 50% rain–snow $T_s$ threshold (°C), $F$ equals snowfall frequency (in this case 0.5, dimensionless), and $a$, $b$, $c$, and $d$ are the fitting parameters (dimensionless). Fitting the curve required a sufficient number of precipitation events per $T_s$ bin, a requirement met by 57.7% of the stations. Observations from stations where a 50% rain–snow $T_s$ threshold could not be computed were still utilized in assessing the meteorological controls on phase partitioning and model optimization as outlined below. Additionally, we computed $T_{90}$ and $T_{10}$ to define the range in which mixed-phase events were probable. In this case, $T_{90}$ and $T_{10}$ corresponded to the temperature at which 90% and 10% of precipitation, respectively, fell as snow.

In order to quantify the effect of RH and $P_s$ on observational snowfall frequency, we divided all precipitation events into six RH classes (40–50%, 50–60%, 60–70%, 70–80%, 80–90%, 90–100%) and four $P_s$ classes (60–70, 70–80, 80–90, 90–105 kPa). RH was calculated per observation using $T_s$, $T_d$, and $P_s$ according to the methods used by Dai[62]. For quality control purposes, we removed observations that had a calculated

RH of less than 10% or greater than 100%, which was 2.6% of the dataset. In addition, 80.5% of the records did not include $P_s$ with observations of $T_s$, $T_d$, and precipitation phase. We therefore used the 1980–2007 average wintertime (December–January–February) $P_s$ from the Modern-Era Retrospective analysis for Research and Applications version 2 (MERRA-2) reanalysis dataset[63,64] for the grid cell (0.5° latitude X 0.625° longitude) in which the station observation was located. For the stations that record $P_s$, the MERRA-2 reanalysis $P_s$ data closely match the station $P_s$ observations (mean bias = 0.11%). After filtering the dataset for the station data analysis, there were a total of 17.8 million precipitation phase observations from 11,924 stations that occurred within the stated $T_s$, RH, and $P_s$ ranges.

**Binary logistic regression phase prediction models.** Given the Boolean nature of classifying precipitation as snow or rain in this study, we optimized three binary logistic regression models on the observed data using combinations of $T_s$, RH, and $P_s$ as predictor variables. Although precipitation falling at $T_s$ near 0 °C can take many forms[37,65], we focused solely on rain and snow as the solid–liquid ratio of mixed-phase events was not reported in the observational dataset. Froidurot et al.[46] noted the efficacy of binary logistic models in discriminating between rain and snow in an analysis of precipitation phase variability in the Swiss Alps. In our study, the models predict the probability of snow occurring (dependent variable), as a function of the independent variables $T_s$ (univariate model), $T_s$ and RH (bivariate model), and $T_s$, RH, and $P_s$ (trivariate model). An event is classified as snow when the probability of snow occurring is greater than or equal to 50% and as rain when less than 50%.

To compare the impact of the three predictor variables on model performance, we optimized three different empirical binary logistic regression models:

Univariate $T_s$ model:

$$p(\text{snow}) = \frac{1}{1 + e^{(\alpha + \beta T_s)}}, \qquad (2)$$

Bivariate $T_s$ and RH model:

$$p(\text{snow}) = \frac{1}{1 + e^{(\alpha + \beta T_s + \gamma \text{RH})}}, \qquad (3)$$

Trivariate $T_s$, RH, and $P_s$ model:

$$p(\text{snow}) = \frac{1}{1 + e^{(\alpha + \beta T_s + \gamma \text{RH} + \lambda P_s)}}, \qquad (4)$$

where $p(\text{snow})$ is the probability of snow occurring (dimensionless), and $\alpha$, $\beta$, $\gamma$, and $\lambda$ are model coefficients (dimensionless). We chose to use an empirical modeling scheme, as opposed to an analytical scheme, given the dataset's inherent spatial and temporal variability, and random errors, as well as a lack of physical information regarding the conditions in the atmospheric column above ~1.5–2 m. To obtain the model coefficients, we ran 250 training simulations using 5000 randomly selected global observations of precipitation phase and the predictor variables. Coefficients were optimized using a generalized linear model in R and Fisher's Scoring Algorithm to reduce model deviance relative to the 5000 random observations. For each of the three model types, we took the mean of the 250 sets of optimized training coefficients to obtain the final model coefficients. To evaluate model skill, we removed the training observations and tested the success rate of each model in predicting precipitation phase within each $T_s$ and RH class in the validation dataset. The success rates for each of the rain–snow $T_s$, $T_w$, and $T_d$ thresholds (Supplementary Table 3) were computed using the same data.

To construct a spatially continuous 50% rain–snow $T_s$ threshold product across the Northern Hemisphere, we applied the optimized bivariate model to the MERRA-2 gridded reanalysis dataset[63,64]. Hourly 2 m $T_s$, specific humidity ($q$), $P_s$, and precipitation data were accessed from 1980 through 2007 and summarized to a daily time step. RH was calculated from the MERRA-2 data using an empirical equation as a function of $q$, $P_s$, and $T_s$. Daily snowfall probability was then simulated for each grid cell using the bivariate model when precipitation was greater than 1 mm and $T_s$ fell within the range of −8 to 8 °C. We then calculated the 50% rain–snow $T_s$ threshold by fitting the hyperbolic tangent to binned estimates of snowfall frequency per MERRA-2 grid cell using Eq. 1.

**Snowfall frequency simulations.** We computed snowfall frequency, the number of snowfall events divided by total precipitation events, using the MERRA-2 reanalysis data outlined above using 18 different precipitation phase methods (Supplementary Table 3) based on values from the literature and the results from the binary logistic phase regression models. Average and standard deviation snowfall frequencies were computed for each grid cell as well as over the Northern Hemisphere as a whole for each of the methods. $T_w$ methods used the $T_w$ values from the MERRA-2 data and $T_d$ was calculated as a function of RH and $T_s$.

**Data availability.** To access the gridded Northern Hemisphere 50% rain–snow $T_s$ threshold product, a formatted version of the observational dataset, and the code used in this manuscript please visit: https://doi.org/10.5061/dryad.c9h35. MERRA-2 reanalysis data were downloaded from the NASA Goddard Earth Sciences Data and Information Services Center (https://disc.sci.gsfc.nasa.gov/). Country outlines for the maps were accessed from Natural Earth (https://naturalearthdata.com).

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

## Acknowledgements

K.S.J. was supported by a NASA Earth and Space Science Fellowship (16-EARTH16F-378). T.S.W. was supported by the United States National Science Foundation Graduate Research Fellowship Program (DGE 1144083). The authors thank Drs. Aiguo Dai, Naoki Mizukami, and Jim Steenburgh for their comments regarding the station precipitation observations and the synoptic-scale processes governing snowfall generation. Publication of this article was funded by the University of Colorado Boulder Libraries Open Access Fund.

## Author contributions

K.S.J., T.S.W., B.L. and N.P.M. designed the study. K.S.J. and T.S.W. performed the analyses and wrote the manuscript. N.P.M. and B.L. provided feedback and edited the manuscript.

## Additional information

**Competing interests:** The authors declare no competing interests.

