## [Peer Review File(PDF 343 kb) · Nature Communications]

Editorial Note: This manuscript has been previously reviewed at another journal that is not operating a transparent peer review scheme. This document only contains reviewer comments and rebuttal letters for versions considered at Nature Communications. Mentions of prior referee reports have been redacted.

Reviewers' comments:

Reviewer #1 (Remarks to the Author):

Review comments for Jennings et al. "Spatial variation of the rain-snow temperature threshold across the Northern Hemisphere"

The rationale driving this paper is that land surface models (LSMs) require realistic information on precipitation phase to generate reliable simulations of surface hydrology, in particular, seasonal snow accumulation and spring runoff. The question arising from this is whether the frequently used assumption of rain/snow separation at 0°C is realistic? To address this, the authors apply a simple bin averaging methodology to derive empirical relationships between the probability of solid/liquid precipitation and air temperature, relative humidity (RH), and air pressure from analysis of surface observations over northern hemisphere land areas. The authors use the temperature threshold where rain and snowfall have an equal 50% probability (T_s) as their metric for characterising precipitation phase. The results provide new insights into the spatial variability of T_s over NH land area, and would be a useful addition to the literature via a publication such as J. Appl. Climate and Meteorology. However, I do not believe the paper scores highly enough on significance and impact for publication in Nature in its current form. The reasons for my conclusion along with some suggestions for improving the paper are outlined below:

1. The focus on the single T_s metric appears to have been made with LSMs in mind where precipitation is either all rain or all snowfall. However, most modern snowpack models in LSMs use mixed phase precipitation input for simulation of snowmelt, metamorphism, and ice layer development. It might therefore be instructive to think about including additional metrics that define the shape of the probability distribution.
2. The paper assumes that LSMs are required to partition total precipitation. This ignores the reality that atmospheric models employ microphysics schemes for diagnosing precipitation type and provide rainfall and snowfall fields for driving hydrological models. Some exploration of the ability of atmospheric and climate models to correctly diagnose precipitation phase would seem to be a logical extension of the empirical analysis. For example, Ikeda et al (2010) showed that high resolution runs of the WRF model were able to provide realistic snowfall simulations over Colorado.
3. The study talks about but does not demonstrate the implications of incorrectly diagnosing precipitation phase on the hemispheric hydrologic system. This is a critical missing element in my opinion. This could be explored through a sensitivity study with a LSM driven by a fixed 0°C rain/snow separation, a run driven with the empirically-derived optimal average " T_s ", and runs driven by precipitation phase diagnosed by various atmospheric models' cloud microphysics schemes. From Figure 3 it looks like the impact is likely to be strongest over semi-arid mountain regions where snow cover is particularly sensitive to warming. In these areas a 0°C rain/snow threshold would seriously underestimate snowfall (and spring runoff). The timing and amount of runoff into the Arctic Ocean may also be sensitive to the parametrization of precipitation phase in LSMs. A clear demonstration of the large scale hydrological impacts of inadequate parametrizations of rain/snow separate would help elevate the significance of the paper.

Minor comments:

1. The period of observational data is not indicated in the Methods.
2. Is observational error taken into account in fitting T50 (eqn. 1)? How large an impact do you think this would have on T50?
3. Figure 1 caption: Are these averages computed over a particular time period? It is not clear from the caption.
4. The discussion under section "Meteorological controls on precipitation phase partitioning" is difficult to follow with all the "bins" the reader has to negotiate. I also find this and the following section to be overly descriptive. Both sections would be considerably improved by a more concise presentation.
5. Some discussion is needed in the introduction on the current ability of atmospheric models at diagnosing precipitation phase.

Ikeda, K., Rasmussen, R., Liu, C., Gochis, D., Yates, D., Chen, F., Tewari, M., Barlage, M., Dudhia, J., Miller, K. and Arsenault, K., 2010. Simulation of seasonal snowfall over Colorado. *Atmospheric Research*, 97(4), pp.462-477.

Reviewer #2 (Remarks to the Author):

This manuscript is novel as this is the first study obtaining a continuous map of rain-snow temperature threshold over the Northern Hemisphere. The manuscript is well written and concise. The authors also conclude that, at near 0°C, the temperature varies significantly with the frequency of snowfall and that the relative humidity also plays an important role in phase partitioning.

I think that the novelty of this manuscript is the continuous map of rain-snow temperature threshold over the Northern Hemisphere. In contrast, as supported by the literature review of the manuscript, the conclusions about the temperature, relative humidity and pressure are not new but it is the first time that it is presented over the entire Northern Hemisphere. The authors, however, needed to use/test these variables to conduct the study.

The paper will interest scientists in many fields that require accurate representation of the precipitation phase at the surface. These are, for example, climate modeling, snowpack and glacier studies, land-surface modeling as well as atmospheric studies. The findings are valuable and set the stage for further scientific investigation and thinking. For example, the relative effects of pressure and relative humidity in given regions could be investigated. The relatively dry environment in the Rocky Mountains leads to a higher temperature threshold but the stations are located at lower pressure values as well (higher elevation), which also lead to higher temperature threshold.

The findings are convincing as they used nearly 18 000 observational data as well as a previously used method (Dai) that classifies the precipitation types. It is not clear, however, if the authors used the amount of precipitation as well. Is the amount of precipitation available in this dataset? I assumed that it is not the case but the sentence on lines 283-285 sounds like the amount of rain and snow are available but not when sleet was reported. It should be clarified in the text. Despite this minor clarification, I believe that no further experiment is needed to make this manuscript acceptable for publication.

Reviewer 1:

1. The focus on the single T_s metric appears to have been made with LSMs in mind where precipitation is either all rain or all snowfall. However, most modern snowpack models in LSMs use mixed phase precipitation input for simulation of snowmelt, metamorphism, and ice layer development. It might therefore be instructive to think about including additional metrics that define the shape of the probability distribution.

Reviewer 1 raises an excellent point, namely that LSMs—as well as hydrologic and snow models—often employ a temperature range in which precipitation can include both rain and snow. This range is typically centered on the 50% rain-snow temperature threshold and cited range values typically vary between 1.0°C and 3.0°C (Feiccabrino et al., 2015; Harpold et al., 2017; Liang et al., 1994; Quick and Pipes, 1977; Wigmosta et al., 1994). Within LSMs and hydrologic models with sufficiently complex snow routines, rainfall and snowfall have different fates within the modeling chain. Therefore it is essential that models not only get the primary phase correct, but also the relative mix of the two. Despite the importance of quantifying the proportions of rain and snow in mixed phase events, relatively little research has been done to provide a physical basis to the range in LSMs. For example, Kienzle (2008) reported a range of 13°C in which both snow and rain were observed in Alberta, Canada, but this study was done based on manual snow depth observations and automated rain measurements without quantification of rain-snow proportions during mixed-phase events. Similarly, Lundquist et al. (2008) noted an approximately 3°C range in which rain and snow could occur in the Sierra Nevada of California. One of the few papers to quantify the mix was Yuter et al. (2006), who used disdrometer data to show there is a mix of rain and snow particles within a 0°C – 1.1°C air temperature range and that the number of rain hydrometeors exceeds those of snow at 0.5°C . However, their study only had one mixed-phase event and was at a single site.

We pursued a binary rain-snow approach in this manuscript because the proportion of mixed-phase precipitation is not provided within the observational dataset. The phase of precipitation is noted by an observer, yet they do not estimate the rain-snow mix, nor would it be straightforward to do so with a simple visual observation. Compounding the problem is the fact that there are WMO precipitation classifications that include rain and/or snow, meaning the code given could be signifying rainfall, snowfall, and/or an unspecified mix of the two (Dai, 2001). We removed measurements with these codes in order to have a higher degree of certainty on whether rain or snow were occurring. Therefore, although the global observational precipitation dataset used in this study offers many unique avenues for analysis, it is not suitable for quantifying the proportion of rain and snow in a mixed-phase event, or even for appropriately identifying a mixed-phase event.

Returning to the reviewers concern, we were indeed interested in whether the shape of the probability curves could be used to estimate more robust rain-snow temperature ranges than the default used by most models. In this context, we have conducted a new

analysis to infer the air temperature at 90% and 10% snow probabilities as a reasonable estimate of the mixed rain-snow temperature range. As noted above, similar approaches have been used by other researchers, but generally only at a single location or within a smaller region (Kienzle, 2008; Lundquist et al., 2008). Our new results show that the temperature range (i.e., the 10% snow frequency temperature minus the 90% snow frequency temperature) varied between 2.5°C and 4.8°C, and the range increased with decreasing relative humidity and surface pressure. In other words, drier and higher locations have a greater range in which both rain and snow are observed. Similar to the findings from our 50% rain-snow air temperature threshold analysis, we found that the temperature range was more sensitive to relative humidity than pressure (i.e., humidity introduces more variation). This new information is included in paragraph 3 of the “*Meteorological controls on precipitation phase partitioning*” section and in Supplementary Table 1.

To respond to Reviewer 1’s point on the shape of the probability curve, we have Supplementary Figure 1 showing observed snow frequency for air temperature, dew point temperature, and wet bulb temperature. We found no fundamental difference in the curve shapes, but we did note a decrease in the 50% rain-snow temperature threshold when using wet bulb (0.3°C) and dew point (-0.3°C) relative to air temperature (1.0°C).

The shortcomings identified in the above paragraphs bring up two important avenues for potential future research:

1. To date, there has been limited work performed on the rain-snow fraction during a mixed-phase event. Future observational work should focus on quantifying the rain-snow mix across sites and climatic conditions. This can be done with a distributed network of disdrometers or high-resolution cameras.
2. As noted above, there is little physical basis to the temperature range employed by many LSMs. In addition to field-based studies, there should be modeling experiments done to test the sensitivity of output to rain-snow temperature ranges, similar to those done by Kienzle (2008) but over a larger spatial scale.

2. The paper assumes that LSMs are required to partition total precipitation. This ignores the reality that atmospheric models employ microphysics schemes for diagnosing precipitation type and provide rainfall and snowfall fields for driving hydrological models. Some exploration of the ability of atmospheric and climate models to correctly diagnose precipitation phase would seem to be a logical extension of the empirical analysis. For example, Ikeda et al (2010) showed that high resolution runs of the WRF model were able to provide realistic snowfall simulations over Colorado.

We agree with Reviewer 1 that atmospheric models resolving cloud microphysics are a promising alternative to the precipitation-partitioning schemes of most land surface models. Such approaches have been used to great effect in CAM, WRF, and for some models in CMIP5, to name just a few examples. Based on the recommendation from Reviewer 1, we have added paragraph 3 to the introduction, citing Ikeda et al. (2010), and noting the important contribution from the atmospheric and climate modeling

communities. We have also added some context for why we chose to focus only on phase prediction schemes using surface data, namely:

1. The relative scarcity of atmospheric measurements compared to ground observations.
2. Model run time and forcing data availability.
3. The popularity of uncoupled land surface models that use surface-based precipitation phase partitioning approaches.

To quantify point 3 above, we performed a Google Scholar and Web of Science search on a selection of LSMs plus hydrologic and snow models that use surface-based phase prediction schemes (table below). The wide availability of forcing data for these models, particularly air temperature, has facilitated their application in diverse regions across the globe. However, the results presented in our manuscript show that a more critical eye must be used when determining how precipitation phase is partitioned when running these models. This is in part where we respectfully disagree with Reviewer 1's suggestion that our work does not have a high enough impact to be published in *Nature Communications*. Given the dramatic spatial variability that we have demonstrated in 50% rain-snow temperature thresholds, we feel that literally hundreds (if not thousands) of modeling projects can be improved through the use of improved thresholds using the proposed phase-partitioning methods.

Cloud microphysics may ultimately be the way forward as computing power improves and higher resolution and globally complete output data become available from atmospheric models. Critically, these models simulate physical relationships between hydrometeors and the atmosphere, not empirical relationships between precipitation phase and surface meteorological measurements. However, at present the wide use of land surface models and the high level of uncertainty introduced by various microphysics schemes within atmospheric models means that surface-based phase prediction methods should be examined more deeply, especially in light of the findings we present in this paper.

Model	Google Citations	WoS Citations
DHSVM	1193	634
NOAH	1630	1169
NOAH-MP	374	248
PRMS	861	NA (technical doc)
SNOWPACK	444	243
Snow17	867	NA (technical doc)
VIC	2215	1133

3. The study talks about but does not demonstrate the implications of incorrectly diagnosing precipitation phase on the hemispheric hydrologic system. This is a critical missing element in my opinion. This could be explored through a sensitivity study with a LSM driven by a fixed 0°C rain/snow separation, a run driven with the empirically-derived optimal average "Ts", and runs driven by precipitation phase diagnosed by various atmospheric models' cloud microphysics schemes. From Figure 3 it looks like the

impact is likely to be strongest over semi-arid mountain regions where snow cover is particularly sensitive to warming. In these areas a 0°C rain/snow threshold would seriously underestimate snowfall (and spring runoff). The timing and amount of runoff into the Arctic Ocean may also be sensitive to the parametrization of precipitation phase in LSMs. A clear demonstration of the large scale hydrological impacts of inadequate parametrizations of rain/snow separate would help elevate the significance of the paper.

To address Reviewer 1's important point without fundamentally changing the scope of the research, we performed a new sensitivity analysis in which we compare the simulated snow frequency from 18 precipitation phase methods over 27 years of MERRA-2 reanalysis data. While this does not quantify the "downstream" effects of misdiagnosing precipitation phase (e.g., soil moisture, streamflow, etc.), it underscores how dramatically the simple choice of precipitation phase algorithm affects model partitioning. We present this information in a new results section ("*Snowfall frequency sensitivity to phase partitioning method*") and a new methods section ("*Snowfall frequency simulations*"), as well as in Figures 5 and 6, and in Supplementary Tables 3 and 4. We also added these additional methods to the "*Simulations of rain-snow partitioning*" results section to show how the incorporation of humidity improves precipitation phase prediction. Figure 4 was updated to reflect these changes/additions (we now display results in a raster format as opposed to a line plot due to overplotting issues).

Overall, these new results show that the selection of phase algorithm generally leads to $\pm 10\%$ uncertainty in snowfall frequency. However, the uncertainty reaches $\pm 30\%$ in semi-arid areas (those identified by Reviewer 1 as likely being the most sensitive) such as the Colorado River Basin as well as the Tibetan Plateau and High Asia. Snow-covered areas in these zones are critical to regional water resources, so it is imperative that hydrologic simulations be as accurate as possible. We also found snowfall frequency uncertainty was largest near average air temperatures of approximately 0.0°C. These findings illustrate that simulations in temperate areas with a rain-snow mix are also sensitive to the selection of precipitation phase algorithm. That such regions are considered most "at risk" to warming-induced snow cover losses (Nolin and Daly, 2006) highlights the importance of our work in detailing the spatial variability of the rain-snow threshold.

In the discussion following these results we present more literature on predicted shifts from snow to rain in cold regions across the globe. We also include new commentary throughout the paper discussing how such shifts are generally associated with reduced snow accumulation, earlier snowmelt, earlier streamflow timing, flashier streamflow, and reduced streamflow efficiency. Therefore, we note there is a large body of literature that details the effect of a shifting precipitation regime from snow to rain with climate warming, while there are relatively few papers detailing the errors introduced by misdiagnosing precipitation phase.

Ultimately, it is outside the scope of this research to run a gridded LSM over the Northern Hemisphere. However, our new sensitivity analysis using a suite of

precipitation phase algorithms increases the impact of our findings on the spatial variability of the rain-snow threshold across the Northern Hemisphere. We hope this research benefits the modeling community and informs scientists that the current approach of many LSMs and other models can result in large errors in the snowfall frequency, in particular for semi-arid areas and sites with winter temperatures near 0°C.

Minor comments:

1. The period of observational data is not indicated in the Methods.

Added to Methods section.

2. Is observational error taken into account in fitting T50 (eqn. 1)? How large an impact do you think this would have on T50?

Error could result from inaccuracies in the meteorological measurements and through the misidentification or improper transcription of precipitation phase. If we assume these errors are random and unbiased, the overall effect should be small based on the extremely large sample size. The lowest RH bin (40–50%) is likely to be the most affected based on the smaller number of observations. We would therefore expect its uncertainty to be greater than the other bins. Although its standard error is indeed greater, its value is still < 0.01%. The low standard error is noted in the Figure 2 caption.

3. Figure 1 caption: Are these averages computed over a particular time period? It is not clear from the caption.

Added to caption for Figures 1 and 2.

4. The discussion under section “Meteorological controls on precipitation phase partitioning” is difficult to follow with all the “bins” the reader has to negotiate. I also find this and the following section to be overly descriptive. Both sections would be considerably improved by a more concise presentation.

Agreed. We removed several sentences and changed the language for clarity in both sections.

5. Some discussion is needed in the introduction on the current ability of atmospheric models at diagnosing precipitation phase.

This along with several other papers were added to the introduction

Reviewer 2:

The findings are convincing as they used nearly 18 000 observational data as well as a previously used method (Dai) that classifies the precipitation types. It is not clear, however, if the authors used the amount of precipitation as well. Is the amount of

precipitation available in this dataset? I assumed that it is not the case but the sentence on lines 283-285 sounds like the amount of rain and snow are available but not when sleet was reported. It should be clarified in the text. Despite this minor clarification, I believe that no further experiment is needed to make this manuscript acceptable for publication.

We revised the text in those lines to make it clear precipitation amounts are not included in the dataset.

We would like to conclude by again thanking you for reviewing our manuscript. We appreciate the time investment you have made as well as the valuable insights provided. If you have any questions regarding the changes we have made or responses given to the reviews, please do not hesitate to contact us.

Best regards,
Keith Jennings and coauthors.

References

- Dai, A., 2001. Global Precipitation and Thunderstorm Frequencies. Part I: Seasonal and Interannual Variations. *J. Clim.* 14, 1092–1111. [https://doi.org/10.1175/1520-0442\(2001\)014<1092:GPATFP>2.0.CO;2](https://doi.org/10.1175/1520-0442(2001)014<1092:GPATFP>2.0.CO;2)
- Feiccabrino, J., Graff, W., Lundberg, A., Sandström, N., Gustafsson, D., 2015. Meteorological Knowledge Useful for the Improvement of Snow Rain Separation in Surface Based Models. *Hydrology* 2, 266–288. <https://doi.org/10.3390/hydrology2040266>
- Harpold, A.A., Kaplan, M., Klos, P.Z., Link, T., McNamara, J.P., Rajagopal, S., Schumer, R., Steele, C.M., 2017. Rain or snow: hydrologic processes, observations, prediction, and research needs. *Hydrol Earth Syst Sci* 21, 1–22.
- Kienzle, S.W., 2008. A new temperature based method to separate rain and snow. *Hydrol. Process.* 22, 5067–5085. <https://doi.org/10.1002/hyp.7131>
- Liang, X., Lettenmaier, D.P., Wood, E.F., Burges, S.J., 1994. A simple hydrologically based model of land surface water and energy fluxes for general circulation models. *J. Geophys. Res. Atmospheres* 99, 14415–14428.
- Lundquist, J.D., Neiman, P.J., Martner, B., White, A.B., Gattas, D.J., Ralph, F.M., 2008. Rain versus Snow in the Sierra Nevada, California: Comparing Doppler Profiling Radar and Surface Observations of Melting Level. *J. Hydrometeorol.* 9, 194–211. <https://doi.org/10.1175/2007JHM853.1>
- Nolin, A.W., Daly, C., 2006. Mapping “at risk” snow in the Pacific Northwest. *J. Hydrometeorol.* 7, 1164–1171.
- Quick, M.C., Pipes, A., 1977. UBC WATERSHED MODEL/Le modèle du bassin versant UCB. *Hydrol. Sci. J.* 22, 153–161.
- Wigmosta, M.S., Vail, L.W., Lettenmaier, D.P., 1994. A distributed hydrology-vegetation model for complex terrain. *Water Resour. Res.* 30, 1665–1679.
- Yuter, S.E., Kingsmill, D.E., Nance, L.B., Löffler-Mang, M., 2006. Observations of Precipitation Size and Fall Speed Characteristics within Coexisting Rain and Wet Snow. *J. Appl. Meteorol. Climatol.* 45, 1450–1464. <https://doi.org/10.1175/JAM2406.1>

Reviewers' Comments:

Reviewer #1:

Remarks to the Author:

I am satisfied with your detailed responses to my comments and the revisions to the m/s. Nice paper plus lots of material for follow-up work! Best regards, Ross Brown

Reviewer #2:

None